# Community Health Nursing Competency and Psychological and Organizational Empowerment of Public Health Nurses: A Cross-Sectional Survey

**DOI:** 10.3390/healthcare9080993

**Published:** 2021-08-04

**Authors:** Ching-Pyng Kuo, Pei-Lun Hsieh, Hsiao-Mei Chen, Shang-Yu Yang, Yu-Ling Hsiao, Shu-Liang Wang

**Affiliations:** 1Department of Nursing, Chung Shan Medical University, Taichung City 40201, Taiwan; pyng@csmu.edu.tw (C.-P.K.); fionajunlom@gmail.com (H.-M.C.); 2Department of Nursing, Chung Shan Medical University Hospital, Taichung City 40201, Taiwan; 3Department of Nursing, College of Health, National Taichung University of Science and Technology, Taichung City 40343, Taiwan; shine@nutc.edu.tw; 4Department of Healthcare Administration, College of Medical and Health Science, Asia University, Taichung City 41354, Taiwan; henry879019@yahoo.com.tw; 5Center of Geriatric Care Resource, Department of Nursing, Fu Jen Catholic University, New Taipei City 242062, Taiwan; 125602@mail.fju.edu.tw

**Keywords:** public health nurse, nursing competence, community empowerment

## Abstract

Aim: This study explored the effect of public health nurses’ current community care nursing competency on the psychological and organizational empowerment of public health services in Taiwan. Design: A cross-sectional nationwide survey design was used. Methods: A self-developed structured questionnaire was administered to public health nurses. They were recruited using a purposive sampling technique, and they participated in community healthcare workshops. Results: The mean score of Community Care Nursing Competence (CCNC) was 3.92 ± 0.83. The mean score in Community Empowerment (CE) was 3.66 ± 0.90. The study revealed that age and communication competence were crucial factors in public health nurses working in the community. With age and through the accumulation of practical experience, public health nurses’ communication competence may also improve, which can further enhance their psychological and organizational empowerment in the nursing workplace.

## 1. Introduction

Promoting community care has been adopted as a policy goal because of the aging population and advancements in medical technology. Community care emphasizes the need to integrate health concepts and healthy lifestyles to achieve a high quality of life, which includes independence, social participation, and dignity [1]. The outbreak of new infectious diseases (such as COVID-19), complications caused by chronic diseases, and decreasing medical budgets have affected global public health [2].

The set of core competencies for public health was devised, comprising seven categories: public health sciences; assessment and analysis; policy and program planning; implementation and evaluation; partnerships, collaboration, and advocacy; diversity and inclusiveness; and communication and leadership [3]. Public healthcare nurses no longer serve as an auxiliary in traditional community healthcare or medical care. Instead, nurses should form partnerships with residents, fully cooperate with the community, perform health assessments and management in the community, and plan health improvement programs and health promotion plans, playing the role of the integrator and coordinator in the community healthcare team [4,5]. 

The importance of encouraging healthcare professionals to be involved in the entire community was emphasized by the COVID-19 epidemic. Educational institutions should adapt their curricula in response to the increasing value attached to global health and community, and public health to address these changes and to cultivate a health workforce with vital competencies and capabilities [2]. Public health nurses’ (PHNs) main responsibility is to implement national public health policies, especially concerning national healthcare, medical care, and disease prevention, which are all within the scope of the major responsibilities of PHNs [6]. Public health nursing practitioners principally work in health centers. The primary services of health centers are associated with the promotion and maintenance of health, disease prevention, and early detection and treatment of diseases, with the aim of ensuring health for all. 

Community care competencies include health promotion and illness prevention, the provision of health education to community residents, and promotion of behavioral changes to enable persons to take control of their health [4,5,7]. Communication competencies include effective written, oral, and electronic communication with clients and interprofessional teams. The development of consultation and advisory skills, and knowledge of languages are also crucial communication competencies [8]. Management competence comprises several facets: employing skills in policy development and program planning; designing, implementing, evaluating, managing, and performing quality assurance on interventions based on the needs of the population; employing problem-solving, critical thinking, and decision-making skills; and performing health assessments of individuals and families during home visits [9].

The definition of empowerment in the literature on community psychology indicates that it can enhance individuals’ competence and self-esteem and can therefore enhance their perception of personal control, which directly affects their health condition. This theory of empowerment can be extended to include forging connections with other people and the community, hoping to obtain more power through the changes in the external environment. Empowerment has a positive impact on productivity in the healthcare. When analyzing nursing productivity in a challenging situation (such as pandemic disease), nurses’ critical thinking, psychological status, and workplace support should be considered as key factors. Studies have demonstrated that empowered nurses more effectively complete their work, display higher organizational productivity, and display more favorable performance in nursing practice [8,10].

The purpose of the study was to identify the current status of the PHNs providing health services in the community and to access the self-perceived community nursing competence, and psychological and organizational empowerment of the PHNs. PHNs’ community nursing competence and empowerment were measured to guide healthcare institution managers in creating a productive and innovative work environment that fosters a sense of empowerment to foster higher-quality outcomes. Specifically, this study addressed the following two research hypothesis: (1) the competencies of care, communication, and management affects CE among PHNs; (2) psychological and organizational empowerment in public health service is related to clinical community health nursing competencies.

## 2. Methods

### 2.1. Study Design

This study adopted a cross-sectional nationwide survey design.

### 2.2. Setting and Samples

The target population of this study was Taiwanese PHNs. PHNs were enrolled from public health centers situated in Taiwan. The nurses worked in public health service-related settings (such as a community healthcare center, public health bureau, or community service center). Nurses working in local medical clinics or hospitals were excluded. 

### 2.3. Measures

Two instruments were self-developed by authors and referenced from the study [11], a community care nursing competence (CCNC) scale, and a community empowerment (CE) scale. A survey was developed by the author to assess factors related to CCNC and CE. 

### 2.4. Community Care Nursing Competence

The three sections of the CCNC scale were CC, communication, and management. The first section of the scale consisted of 15 items rated using a five-point Likert scale. The second and third sections of the scale measured communication competence and management competence, each section included eight items rated on a five-point Likert scale, with the responses ranging from *strongly disagree* to *strongly agree*. Higher scores indicated higher care competence in providing community health services. 

### 2.5. Community Empowerment

The two sections of the CE scale were psychological CE (PCE) and organizational CE (OCE). The first section of the scale comprised 10 items to assess psychological empowerment in PHNs and was rated using a five-point Likert scale. The second section of the scale measured empowerment in the working organization and comprised 15 items rated on a five-point Likert scale. Higher scores indicated higher community empowerment for providing community health services.

### 2.6. Ethics Approval and Consent to Participate

This study has been approved by the Ethical Committee of China Medical University, Taichung, Taiwan, for Research Data on 11 November 2019 (decision number CRREC-108-125). The researchers explained the research purpose, process, and protection of personal rights to the participants, and informed consent forms were provided before data collection. Written information about the study, including the participants’ legal rights regarding participation and confidentiality, was provided. The participants were assured that it was voluntary to participate in the study and that they were free to withdraw from the study at any time. 

### 2.7. Data Collection

Between December 2019 and March 2020, 244 paper questionnaires were distributed to public health nurses. These public nurses had completed community care related on-the-job training, and the questionnaire survey was conducted. A total of 197 valid questionnaires were returned. Therefore, the response rate of 80.74%.

### 2.8. Data Analysis

Descriptive statistics were used to describe the major study variables and sample demographics. A one-way analysis of variance and *t* tests were used to analyze the variance among the demographic data, CCNC, and CE. The bivariate Pearson correlation measures were used to direction of linear relationships between pairs of continuous variables. Fisher’s least significant difference (LSD) method was used in ANOVA to create confidence intervals for all pairwise differences between factor level means. Furthermore, stepwise regression was used to test two of the hypotheses and to predict the significant factors affecting the CE of PHNs.

## 3. Results

### 3.1. Research Subject Demographic Information

Most research subjects were female, aged between 40 and 49 years (*n* = 83, 42.10%), and 72.60% were married (*n* = 143). Most participants (*n* = 133, 67.50%) had a baccalaureate degree, and 13.70% had a master’s degree or higher. The occupation title of most of the participants was “registered nurse” (*n* = 137, 69.60%). Most subjects worked in public health centers (*n* = 176, 89.30%). Most respondents had over 10 years of experience in public health nursing (*n* = 61, 31.0%) (Table 1).

### 3.2. Community Care Nursing Competence of Public Health Nurses

The content validity index values for the CCNC scale was 0.90. CCNC involves the aspects of CC, communication, and management (shown in Figure 1). The total scale of this study displayed acceptable internal consistency (Cronbach’s α = 0.98). The average score on the overall CCNC was 3.92 ± 0.83, which is between *neutral* and *agree*. Among the dimensions, “Communication” (mean = 4.05 ± 0.78) displayed the highest score, followed by “CC” (mean = 4.03 ± 0.90) and “Management” (mean = 3.98 ± 0.81). In the “CC” dimension, “Provide health check and early screening services for related chronic diseases” scored the highest (mean = 4.59 ± 0.63), followed by “Provide blood pressure, blood glucose, and cholesterol measurement services” (mean = 4.4 ± 0.86). In the dimension of “Communication”, “Maintain effective communication with the client and listen and accept client concerns” scored the highest (mean = 4.15 ± 0.80), followed by “Provide the clients with proper explanations and descriptions when implementing related measures or care plans” (mean = 4.12 ± 0.80) and “Observe and use nonverbal communication skills to establish high-quality nurse–patient relationships” (mean = 4.12 ± 0.80). In “Management”, “Cooperate with central government policies to implement chronic disease–related care” scored the highest (mean = 4.19 ± 0.76), followed by “Cooperate with the organizational departments (such as long-term care and social welfare)” (mean = 4.10 ± 0.75). 

### 3.3. Community Empowerment of Public Health Nurses

The content validity index values for the CE scale was 0.92. The mean overall CE score was 3.66 ± 0.90. The “Psychological Empowerment Scale” dimension scored the highest (mean = 3.91 ± 0.74), followed by “The work performed is crucial for health promotion” (mean = 4.20 ± 0.67) and “The work performed is critical for promoting community health” (mean = 4.19 ± 0.70); the items with the lowest scores were “I am highly proficient in the skills required at work” (mean = 3.74 ± 0.70) and “I can influence what happens within the work unit” (mean = 3.74 ± 0.75). The mean score on the subscale “Organizational Empowerment” was 3.79 ± 0.75, and among the items, “I can satisfy the work requirements and complete the work as scheduled” scored the highest (mean = 3.95 ± 0.64), followed by “The budget is sufficient for the work that must be performed” (mean = 3.91 ± 0.71); the items scoring the lowest were “The human resources (both inside and outside the organization) required to perform the work are provided” (mean = 3.63 ± 0.90) and “I receive sufficient positive encouragement from the supervisor” (mean = 3.65 ± 0.89). 

### 3.4. The Correlation between the Personal Attributes, Community Care Competence, and Community Empowerment of Public Health Nurses

The analysis of the correlation between personal attributes and CCNC revealed that the service unit (F = 1.936, *p* = 0.001) was correlated with the CCNC. The CCNC of nurses working in health centers was higher than that of those working in health bureaus and other CC locations. The correlation analysis of the personal attributes and CE of the PHNs demonstrated that age (F = 2.179, *p* = 0.015) was correlated with CE. Pearson’s correlation coefficient was used to analyze the correlation between the CC competence and the CE of the nurses (see Table 2 for details).

### 3.5. Factors Affecting Community Empowerment

Multiple regression analysis was used to determine factors affecting the community empowerment by including variables with statistical significance. The results (Table 3) revealed that the major predictors were “Age” (B = 0.18, *p* = 0.021) and “Communication Competence” (B = 0.17, *p* = 0.002), and the explanatory power of the community empowerment was 28 per cent. 

## 4. Discussion

The research results showed that “Age” was one of the factors affecting community empowerment. The largest proportion of participants 31% had more than 10 years of service in public health units, 23.4% had between 5 and 10 years, and 28.4% had less than 5 years. Approximately 60% of participants had a bachelor’s degree. These findings indicate that nursing staff in public health services should be equipped with sufficient field experience to be familiar with conditions in the local community and to establish partnerships [12]. Approximately 40% of the participants did not have a bachelor’s degree, which may result in a relatively weak perception of empowerment among the PHNs, both psychologically and concerning the working environment [6,11].

A significant positive correlation was observed between the degree of chronic disease care implementation and the perception of empowerment, indicating that more frequently performing CC was associated with a higher score on relative empowerment perception, which is in accordance with the results from numerous studies, including a study on empowering PHNs in the care of clients and improving PHNs’ self-efficacy [1,5,13], a study on the effect of psychological empowerment on CC competence [11], and a study on the positive effects of self-efficacy on work performance [6,9]. The results of the present study indicated that PHNs’ competence in the implementation of CC management and the degree of implementation increased when their perception of psychological empowerment and empowerment in the workplace was enhanced.

The CC results indicated that health checks and early screenings of related chronic diseases were the services most commonly provided by PHNs, suggesting that the PHNs’ care services were primarily focused on preventive health services. Studies have reported that PHNs provide services for community health promotion and preventive care services and that the frequency of providing chronic disease care services is the highest [1,4,7]. However, the participants reported that their competencies were insufficient when providing individual care plans based on client needs and preventing comorbidities, which may be related to PHNs’ experience in the care of disease [5,8]. When the COVID-19 pandemic occurred in 2020, communities faced a tremendous public health threat. Therefore, PHNs should exercise caution when visiting clients in the community and should increase their relevant knowledge of emerging infectious disease and prevention measures. Furthermore, PHNs should also instruct the public on disease prevention and should provide referrals to competent professionals to control hazardous public health situations [1,7]. 

“Communication” items displayed the highest scores of CCNC, indicating that PHNs should be equipped with strong communication competencies because these enable them to establish nurse–patient relationships and to understand clients concerns when providing care plans; this finding accorded with results reported in the literature [4,5,14]. PHNs could “Maintain effective communication with the client and listen and accept client concerns”, “Provide the clients with proper explanations and descriptions when implementing related measures or care plans”, and “Observe and use nonverbal communication skills to establish high-quality nurse–patient relationships”, indicating that communication plays a crucial role in CC services. The results of this study demonstrated that communication competence affects nurses’ perception of empowerment, a finding that is in accordance with the findings in the literature [5,7]. However, the nurses perceived personal insufficiencies in the items of “Make decisions concerning treatment and care plans with community clients” and “Use community resources to achieve various treatments or health promotion”, demonstrating that nurses must improve communication with the clients regarding their needs when discussing care plans [4,11]. 

The CE results indicated that the PHNs believed that “The work that is performed is critical for health promotion” and “The work that is performed is crucial in promoting community health”, and these beliefs were associated with psychological empowerment. Regarding organizational empowerment, the participants with a self-perception of “I can meet the work requirements and complete the work as scheduled” could also perceive that they were empowered in the workplace, which improved their work efficiency [15]. Certain aspects of psychological empowerment, such as the sense of self-meaning of work, care management competence, and decision-making related participation, can be enhanced with related resources, such as more work-related information, specific suggestions for information, addressing problems, and positive encouragement, all of which can enable PHNs to be independent and to leverage all community resources to manage chronic diseases. PHNs would thus be able to provide patients and their family members with relevant chronic disease care information to address their concerns and to implement strategies with partner organizations to improve the quality of care for clients with chronic disease in the community. 

Professional competence is crucial in providing quality healthcare services. Quality of care requires that nursing staff members possess the competencies needed to satisfy complex healthcare demands. Internationally, studies have indicated that higher staffing a higher number of nurses in general healthcare are associated with a higher quality of care, improved patient outcomes, and fewer adverse events. 

## 5. Limitations

This study adopted a cross-sectional correlation design; thus, the results depended on the condition and status of the respondents at the time of completing the questionnaire. The respondents completed their statements subjectively, making it difficult to assess their actual intention. Moreover, this study was a quantitative study, which made it difficult to understand the PHNs’ CCNC and empowerment in community. 

## 6. Conclusions

This study determined that, in addition to basic competencies in general chronic disease care management, PHNs in the community should gain experience with clients with chronic diseases or residents in the community so that the PHNs’ communication capabilities can be improved for them to share their experiences. PHNs are poised to lead advancements in public health and healthcare, especially in terms of solving health inequities. PHNs with a bachelor’s degree or higher are equipped to handle numerous determinants of health and to fully participate in the challenges of achieving and maintaining public health. The scope of their responsibilities include community-building, health promotion, policy reform, and implementing system-level changes to promote and protect public health. PHNs, as the leaders in the improvement of health and the promotion of health equality, play a crucial role in the future of healthcare.

## Figures and Tables

**Figure 1 healthcare-09-00993-f001:**
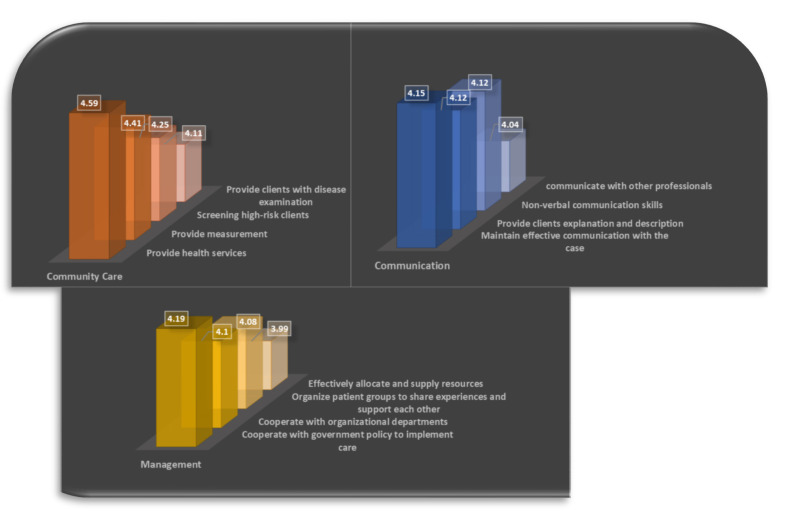
Analysis of community care nursing competence (*n* = 197).

**Table 1 healthcare-09-00993-t001:** Demographic characteristics of the subjects (*n* = 197).

Variables	Categories	*n*	(%)
Gender			
	Male	7	3.56
	Female	190	96.44
Age (years)			
	20–29	20	10.20
	30–39	71	36.00
	40–49	83	42.10
	>50	23	11.70
Marital			
	Single	49	24.90
	Married	143	72.60
	Divorce	4	2.00
	Missing	1	0.50
Education			
	Junior college	37	18.80
	Baccalaureate	133	67.50
	Graduate and above	27	13.70
Workplace			
	Public Health Bureau	10	5.10
	Public Health Center	176	89.30
	Others (ex: Community Service Center...)	5	2.50
	Missing	6	3.10
Work position			
	Registered nurse (Public Health Nurse)	137	69.60
	Head nurse	31	15.70
	Others (Government employee, working in Public Health Bureau, Public Health Center, Community Service Center)	27	13.60
	Missing	2	1.10
Experience in public health (year)			
	<1	29	14.70
	>1–5	56	28.40
	>5–10	46	23.40
	>10	61	31.00
	Missing	5	2.50

**Table 2 healthcare-09-00993-t002:** Correlation between community care nursing competence and community empowerment (*n* = 197).

	CCC-T	CC	C	M	(CE-T)	PCE	OCE
Community Care Nursing Competence Total score (CCNC-T)	1						
Community Care (CC)	0.97 (<0.000) ***	1					
Communication (C)	0.94 (<0.000) ***	0.84 (<0.000) ***	1				
Management (M)	0.94 (<0.000) ***	0.85 (<0.000) ***	0.88 (<0.000) **	1			
Community Empowerment Total score (CE-T)	0.23 (0.002) **	0.21 (0.004) **	0.23 (0.001) **	0.21 (0.003) **	1		
Psychological Community Empowerment (PCE)	0.21 (0.003) **	0.20 (0.006) **	0.22 (0.002) **	0.19 (0.007) **	0.20 (<0.000) ***	1	
Organizational Community Empowerment (OCE)	0.22 (0.002) **	0.20 (0.004) **	0.23 (0.001) **	0.21 (0.003) **	0.98 (<0.000) ***	0.82 (<0.000) ***	1

** Significant at *p* < 0.01, *** Significant at *p* < 0.001, Statistics are based on Pearson correlation analysis.

**Table 3 healthcare-09-00993-t003:** Factors affecting community empowerment (*n* = 197).

				Unstandardized Coefficients		Standardized Coefficients		95% Confidence Interval	
**Model**	**R**	**R Square**	**Adjusted *R* Square**	**Β**	**SEB**	**β**	**t (*p*)**	**Up**	**Low**
	0.28	0.08	0.07						
(Constant)				2.28	0.35		6.50		
Age				0.18	0.08	0.16	2.33(0.021) *	0.07	0.28
Communication Competence				0.17	0.08	0.16	2.29(0.002) **	0.13	0.31

* Significant at *p* < 0.05. ** Significant at *p* < 0.01.

## Data Availability

The data presented in this study are available on request from the corresponding author.

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
