# Peer review of "Community Health Nursing Competency and Psychological and Organizational Empowerment of Public Health Nurses: A Cross-Sectional Survey"

_healthcare, 2021, doi:10.3390/healthcare9080993_

Round 1

Reviewer 1 Report

The authors present results from a survey measuring competence and empowerment among 197 public health nurses in Taiwan.  

Overall impression: This is certainly a highly relevant topic, which should be published. Yet, I found it difficult to follow the argumentation of the authors throughout the manuscript. The research goal is unclear and the presentation of methods and results lacks relevant details and clarity. I am afraid that I could not assess the quality of the paper until these general issues are solved. The authors should use a comparable paper from a high-impact journal as example and follow the STROBE guideline on cross-sectional studies for improving reporting. 

At the moment, I could just add some suggestions for a general revision of the manuscript.  

Introduction: 

  1. Short description of theoretical background from general to a specific topic for the paper. The scope is much to general at the moment.
  2. Discuss the current state of research on the specific topic - both from an international angle and based on studies in your sample region. What were open points that your study will address? What was the specific motivation for your study (maybe a serious problem in care identified previously)? If applicable please also state a research question and hypothesis. At the moment, the stated goal is "to understand" and "to explore", which seems way to general for a single paper. 

Methods

  1. Remove research goals (like in 2.1) and findings (like presenting Cronbach's alpha) in the methods section. This is either part of the introduction or results section. Describe all statistical methods to assess the reliability and validity of the survey and present the results of that in the results section.
  2. The estimation of the minimum sample size makes only sense, if there is a clear outcome to be compared. In this more exploratory study, such a calculation is unnecessary and misleading.
  3. Please explain the development of the survey in more detail. 
  4. The statistical analysis needs also much more details. Which "multiple regression analysis" presented in 3.5 was used specifically?
  5. In could not follow how the methods used help to fulfil the goal of the paper. For instances, the bivariate correlations among the dimensions are less informative.#

Results

  1. Don't just describe all the different items and their mean scores (means are often less informative for Likert scales). Present the results for your specific research goal. For instances, table 2 could be moved to the web supplement of the paper. Rather present a barchart or ranking of interesting aspects.
  2. The correlation analysis does not provide interesting aspects and is too descriptive (could also be moved to the web supplement). Perhaps focus on a multivariate model, where the goal is to explain CE or CC based on all the variables you have and present that as forrest plot or table including confidence intervals.

Discussion

1. Please follow the general structure of a discussion: Restate briefly research question and method to answer it, mention your most relevant findings to the question, say what is new about that or what it adds to the knowledge and mention the merit of your research. Then discuss single striking findings in the light of earlier research on each aspect and finally explain the limitations in more detail and how each limitation may had affected your findings. 

Finally, the conclusion and abstract should be adapted to the more structured article. 

Author Response

Dear reviewer :

Thank you for your kind letter dated June 28, 2021. We find your comments very important. We believe that your suggestions and editing will greatly improve the quality of our paper. We have, hence, revised our manuscript as you advised and marked our corrections in the  following chart, and carefully proof-read the manuscript to minimize typographical, grammatical, and bibliographical errors.

Sincerely,

Pei-Lun Hsieh

Assistant Professor

Department of Nursing, College of Health, National Taichung University of Science and Technology, Taichung, Taiwan R.O.C.

No. 193, Sec. 1, San-Min Rd., Taichung, Taiwan, (40343), R.O.C.

Telephone: +8864-22196854  

Email Address: [email protected]

Reviewer 2 Report

Just 3 minor English improvements I found, interesting article overall!

a questionnaire is singular.  therefore you must use, a questionnaire was administered in the abstract.

      A self-developed structured questionnaire were administered to Taiwanese public health nurses, recruited using a purposive sampling technique, who participated in community health care workshops.      A self-developed structured questionnaire was administered to Taiwanese public health nurses. They were recruited using a purposive sampling technique, and they participated in community health care workshops.       

This study adopted a cross-sectional nationwide survey design to investigate the current state of community health nursing competence, and psychological and organizational empowerment among PHNs 

    I don’t like how the use of two ands are used and work to restructure the sentence.      

This study determined that in addition to basic competencies in general chronic dis- 292 ease care management, PHNs in the community should gain experience with clients with 293 chronic diseases or residents in the community to improve their communication and to 294 enable them to share their experiences.  

    Confusing sentence.    Who needs to improve communication?  The patient or the PHNs? Clarify in the sentence a bit better.

Author Response

Dear reviewer :

Thank you for your kind letter dated June 17, 2021. We find your comments very important. We believe that your suggestions and editing will greatly improve the quality of our paper. We have, hence, revised our manuscript as you advised and marked our corrections in the  following chart, and carefully proof-read the manuscript to minimize typographical, grammatical, and bibliographical errors.

Sincerely,

Pei-Lun Hsieh

Assistant Professor

Department of Nursing, College of Health, National Taichung University of Science and Technology, Taichung, Taiwan R.O.C.

No. 193, Sec. 1, San-Min Rd., Taichung, Taiwan, (40343), R.O.C.

Telephone: +8864-22196854  

Email Address: [email protected]

Reviewer 3 Report

This paper studies community health nursing competence in Taiwan. The research methods and conclusions are convincing, but the paper is too technical and descriptive. In order to make the paper more academic, I have the following suggestions:

1. The author may consider establishing a theoretical framework and an analytical framework for the paper (however, not mandatory).

2. The author can alternatively expand on the existing framework. For instance, in the section of introduction, the "empowerment approach" (page 2) can be introduced with more details, and at the end of the paper the innovation and theoretical contribution of this paper should be more elaborated.

3. At least this paper can do more literature review, the current use of academic literature is too few.

4. It is suggested that the author can address the problem of data representativeness with a few sentences.

5. A comprehensive proofreading by a native English speaker for text editing and proofreading is recommended.

Author Response

Dear reviewer :

Thank you for your kind letter dated July 04, 2021. We find your comments very important. We believe that your suggestions and editing will greatly improve the quality of our paper. We have, hence, revised our manuscript as you advised and marked our corrections in the  following chart, and carefully proof-read the manuscript to minimize typographical, grammatical, and bibliographical errors.

Sincerely,

Pei-Lun Hsieh

Assistant Professor

Department of Nursing, College of Health, National Taichung University of Science and Technology, Taichung, Taiwan R.O.C.

No. 193, Sec. 1, San-Min Rd., Taichung, Taiwan, (40343), R.O.C.

Telephone: +8864-22196854  

Email Address: [email protected]

Round 2

Reviewer 1 Report

This study aims to assess the association of nursing competence and organizational empowerment in a community care setting in Taiwan. The cross-sectional study is based on a survey among 244 public health nurses (response rate 80.8%). Self-assessed nursing competence was measured along three dimensions community care, communication and management, while empowerment was divided into psychological and organizational aspects. Among the dimensions of nursing competence (overall score 3.92), communication seemed to exhibit the highest average score, while management scored lowest. Community empowerment scored 3.66 on average, whereby the psychological dimension seemed to be higher than the organizational one.  The bivariate correlations between empowerment and nursing competence was 0.23 ranging between 0.19 and 0.23 for the different dimensions. In the multivariate regression model, age and communication seemed to explain most of the variation in empowerment.

General comment: The paper has improved compared to the initial submission. Yet, I am afraid that the study is still not presented in a way that should be published in an international journal. The research question is still unspecific and is not adequately motivated by the current state of other work in the field and research gaps (highlighted by the fact that there are only few references to other work in total). Despite stating that the authors followed the STROBE guidelines, the paper does not follow them strictly. The tables and figures are not self-explaining and do not meet requirements for publications. The cohort, surveys and statistical methods of the paper are still insufficiently described, particularly how the independent predictors of empowerment were identified, which is the most relevant part of the paper. The presentation of bivariate correlations does not provide interesting evidence for the research goal. Introduction and discussion do not follow a clear structure complicating the understanding of the research project. Finally, such a complex survey study should provide extensive material as online supplement so that the analyses of the paper are transparent and comprehensible.

Reviewer 3 Report

The quality of this paper has been improved, it still needs some work of editing and final proofreading.